# Barriers and facilitators to prudent antibiotic prescribing for acute respiratory tract infections: A qualitative study with general practitioners in Malta

Erika A. Saliba-Gustafsson[1,2]*, Anna Nyberg[3], Michael A. Borg[4,5], Senia Rosales-Klintz[1,6], Cecilia Stålsby Lundborg[1]

1 Department of Global Public Health, Health Systems and Policy (HSP): Improving Use of Medicines, Karolinska Institutet, Stockholm, Sweden, 2 Division of Primary Care and Population Health, Stanford University School of Medicine, Stanford, California, United States of America, 3 Faculty of Social Sciences, Stockholm University, Stockholm, Sweden, 4 Department of Infection Prevention and Control, Mater Dei Hospital, Msida, Malta, 5 Faculty of Medicine and Surgery, University of Malta, Msida, Malta, 6 Unit of Surveillance and Response Support (SRS), European Centre for Disease Prevention and Control, Solna, Sweden

* erika.saliba@gmail.com

**Data Availability Statement:** All relevant data are presented in the paper. Since we present interview data, some information disclosed by general

## Abstract

### Background

Antibiotic resistance is a leading global public health concern and antibiotic use is a key driver. Effective interventions are needed to target key stakeholders, including general practitioners (GPs). In Malta, little is known about factors that influence GPs' antibiotic prescribing, making it challenging to implement targeted interventions. We therefore aimed to explore GPs' understanding of antibiotic use and resistance, and describe their perceived barriers and facilitators to prudent antibiotic prescribing for acute respiratory tract infections in Malta.

### Methods

Face-to-face individual semi-structured interviews were held with a quota sample of 20 GPs in 2014. Interviews were audio recorded and transcribed verbatim, and later analysed iteratively using manifest and latent content analysis. Findings were collated in a socioecological model to depict how GPs as individuals are embedded within larger social systems and contexts, and how each component within this system impacts their prescribing behaviour.

### Findings

We found that GPs' antibiotic prescribing decisions are complex and impacted by numerous barriers and facilitators at the individual, interpersonal, organisational, community, and public policy level. Predominant factors found to impact GPs' antibiotic prescribing included not only intrinsic GP factors such as knowledge, awareness, experience, and misconceptions, but also several external factors. At the interpersonal level, GPs' perceived patient demand

practitioners may make them identifiable, particularly since the study has been carried out in a small country. Providing full transcripts may compromise their identity and that of others, going against ethical considerations. All data are currently archived at the department for a minimum of 10 years as per the University's archiving policy. Our research group administrator, who is not a co-author on this paper, would be able to assist with any future data access requests. Her full name is: Vijaylakshmi Prabhu and her email address is: vijaylakshmi.prabhu@ki.se. Alternatively, one may directly contact the archivist using the following email address: registrator.gph@ki.se.

**Funding:** This work was supported by Karolinska Institutet funding for doctoral education (KID-funding 3–1233/2013). It was also supported by funding available to CSL at Karolinska Institutet. This study received no other specific grant from any public, commercial or not-for-profit funding agencies. Funders had no role in study design, data collection and analysis, decision to publish, or preparation of the manuscript.

**Competing interests:** I have read the journal's policy and the authors of this manuscript have the following competing interests: At the time of the study, SRK was employed at Karolinska Institutet, Sweden. She is currently employed by the European Centre for Disease Prevention and Control (ECDC). The views and opinions expressed herein are the authors' own and do not necessarily state or reflect those of ECDC. ECDC is not responsible for the data and information collation and analysis and cannot be held liable for conclusions or opinions drawn. EASG, AN, MAB and CSL have no competing interests to declare.

and behaviour to be a persistent issue that impacts their prescribing decisions. Similarly, some GPs found pressure from drug reps to be concerning despite being considered an important source of information. Organisational and public policy-level issues such as lack of access to relevant antibiotic prescribing guidelines and current antibiotic resistance data from the community, were also considered major barriers to appropriate antibiotic prescribing. Utilisation of diagnostic testing was found to be low and GPs' perceptions on the introduction of rapid point-of-care tests to support antibiotic prescription decisions, were mixed.

## Conclusion

This study revealed the complexity of the antibiotic prescribing decision and the numerous barriers and facilitators that impact it, visualised through a socioecological model. Addressing GPs' antibiotic prescribing practices will require targeted and coordinated implementation activities at all levels to change behaviour and address misconceptions, whilst also improving the physical and social environment.

## Trial registration number

NCT03218930; https://clinicaltrials.gov/ct2/show/NCT03218930.

## Introduction

Antibiotic resistance (ABR) is a multi-sectoral challenge of leading global public health concern that threatens patient safety [1]. In Europe, an estimated 33,000 people die of antibiotic-resistant infections annually, with patients under 1 and over 64 years bearing the highest burden [2]. Antibiotic consumption is one of the main drivers of ABR [3–5], and the correlation is even stronger in southern European countries where consumption rates are highest [3].

In Europe, most antibiotics prescribed for systemic use are provided to patients in the community setting [6]. Acute respiratory tract infections (ARTIs) remain the most common indication, despite their self-limiting nature [5]. Indeed, antibiotics confer little benefit for most ARTIs and the risk for complications when withholding antibiotic treatment is minimal [7–11]. Studies have revealed a plethora of factors that influence general practitioners' (GPs') antibiotic prescribing, including clinical, patient, environmental and cultural factors, patient demand and expectations, and provider attitudes and characteristics [12–15]. Despite the complexity, GPs often diagnose infections based on clinical features, prescribing antibiotics empirically without using rapid point-of-care tests (POCTs) [16]. This places uncertainty in GPs' decision-making, who may prescribe antibiotics to be on the safe side.

Several antibiotic stewardship programmes and public campaigns have been implemented in the community to improve antibiotic prescribing for ARTIs, with varying results [17–22]. However, behavioural interventions in high-consuming southern European countries remain scarce. Since 2013, Malta, a southern European country, has reported among the highest antibiotic consumption rates in Europe [23–25]. In 2018, 42% of Maltese respondents claimed to have consumed at least one antibiotic course during the past calendar year (EU-average: 32%); almost all (96%) were prescribed by a medical doctor [23]. The top two reasons were sore throat (22%) and flu (14%) [23]. Similarly, we showed that 45.7% of patients with respiratory tract complaints were prescribed antibiotics by a GP between 2015 and 2016 [26]. Since the majority of antibiotic consumption in Malta occurs in the community [27], and most antibiotics are

obtained through a medical prescription [23], identifying barriers and facilitators to GPs' antibiotic prescribing is crucial to implement effective and targeted interventions. We therefore aimed to explore GPs' understanding of antibiotic use and resistance, and describe their perceived barriers and facilitators to prudent antibiotic prescribing for ARTIs in Malta.

## Methods

This study was part of the Maltese Antibiotic Stewardship Programme in the Community (MASPIC). The project aimed to improve GPs' antibiotic prescribing behaviour for acute respiratory tract complaints in Malta through a tailored, multifaceted social marketing intervention [28].

### Setting

In Malta, patients are not registered to a particular GP and are free to choose their own doctor. About two-thirds of primary care is provided by private sector GPs, primarily solo practitioners, and home visits are still in demand [29,30]. Many private GPs practice within consultation rooms situated in retail pharmacies, and patients pay out-of-pocket; no subsidy or reimbursement applies. In the public sector, GP clinics are walk-in clinics and services are free at point-of-delivery to all citizens. Antibiotics are prescription-only medicines by law, purchased out-of-pocket from private community pharmacies [31].

### Study design and participants

A qualitative methodology was deemed appropriate to address our research question. Following a literature search, a semi-structured interview guide (S1 File) was developed and adapted to the local context. It was pilot tested with 6 GPs in March 2014 and revised accordingly. Registered GPs were eligible to participate if they worked part-time or full-time, and in the public and/or private sectors. Using quota sampling [32], all eligible GPs with an available phone number were listed and divided into strata based on: (i) years of experience, (ii) sex, and (iii) locality of residence. Thirty GPs were contacted, however 4 were ineligible and 6 declined participation. Ultimately 20 GPs participated (Table 1). Age ranged from 32–70 years (mean = 52 years). GP experience ranged from 7–45 years (mean = 26 years).

### Data collection

Individual semi-structured interviews were held by the first author (EASG), a trained qualitative researcher, between August and September 2014. GPs were interviewed at locations convenient to them, often their own clinic outside patient hours. Interviews lasted 25 to 67 minutes (median = 40 minutes) and data were collected until saturation was reached. Interviews were audio recorded and transcribed verbatim.

### Data analysis

EASG and AN (an experienced qualitative researcher) analysed interview transcripts iteratively using manifest and latent content analysis with an inductive approach [33,34]. Briefly, transcripts were read independently numerous times to get a sense of the whole. Meaning units were identified and abridged into condensed meaning units, from which codes were derived. Similar codes were grouped into sub-categories and coalesced into categories, then grouped into sub-themes to extract an overarching theme. Analysis was discussed repeatedly between EASG and AN, and later with all co-authors, until consensus was reached. Finally, findings were visualised in a socioecological model as described by McLeroy et al. [35], to

**Table 1. General practitioners' demographic characteristics (n = 20).**

|  |  | Frequency, *n* |
|---|---|---|
| **Sex** | Male | 14 |
|  | Female | 6 |
| **Age (years)** | 30–39 | 2 |
|  | 40–49 | 4 |
|  | 50–59 | 11 |
|  | 60–69 | 2 |
|  | 70–79 | 1 |
| **Experience in general practice (years)** | 0–9 | 2 |
|  | 10–19 | 3 |
|  | 20–29 | 10 |
|  | 30–39 | 3 |
|  | 40–49 | 2 |
| **Health sector of practice** | Public | 4 |
|  | Private | 14 |
|  | both | 2 |

depict how GPs' behaviour is influenced by various processes and interactions at 5 levels: individual, interpersonal, organisational, community, and public policy.

## Ethical considerations

Ethical approval was sought from the University of Malta Research Ethics Committee but deemed exempt. Nonetheless, standard ethical considerations were adhered to. Participants were informed verbally and in writing about the study's aim and their role, and signed informed consent was obtained. Interviewees participated voluntarily and were free to withdraw from the study without consequence.

## Findings

Our findings revealed one overarching theme, **GPs' antibiotic prescribing decisions are complex and impacted by numerous barriers and facilitators at the individual, interpersonal, organisational, community, and public policy level,** and are presented over four sub-themes, supported by illustrative quotes. We found that GPs' antibiotic prescribing is impacted by both intrinsic and extrinsic factors, including patient demand and behaviour, information from drug reps, communication and collaboration with other healthcare professionals, availability of guidelines, access to diagnostic testing, and ease of referral. This reflects how GPs as individuals are embedded within larger social systems and contexts, and how each component within this system impacts their prescribing. We therefore collated our findings into a socioecological model to provide an illustrative overview of barriers and facilitators impacting GPs' antibiotic prescribing at all levels of the system (Fig 1).

## Sub-theme I: GPs' views on antibiotic use and resistance, and the management of ARTIs

**Category I: GPs' views on antibiotic use, ABR, and the pathogenesis of ARTIs.** Antibiotics were viewed as precious tools that must be used judiciously. GPs acknowledged that antibiotics should not be overprescribed but feared that delaying necessary antibiotic treatment could result in more complicated infections. Nevertheless, a "wait-and-see approach", i.e.,

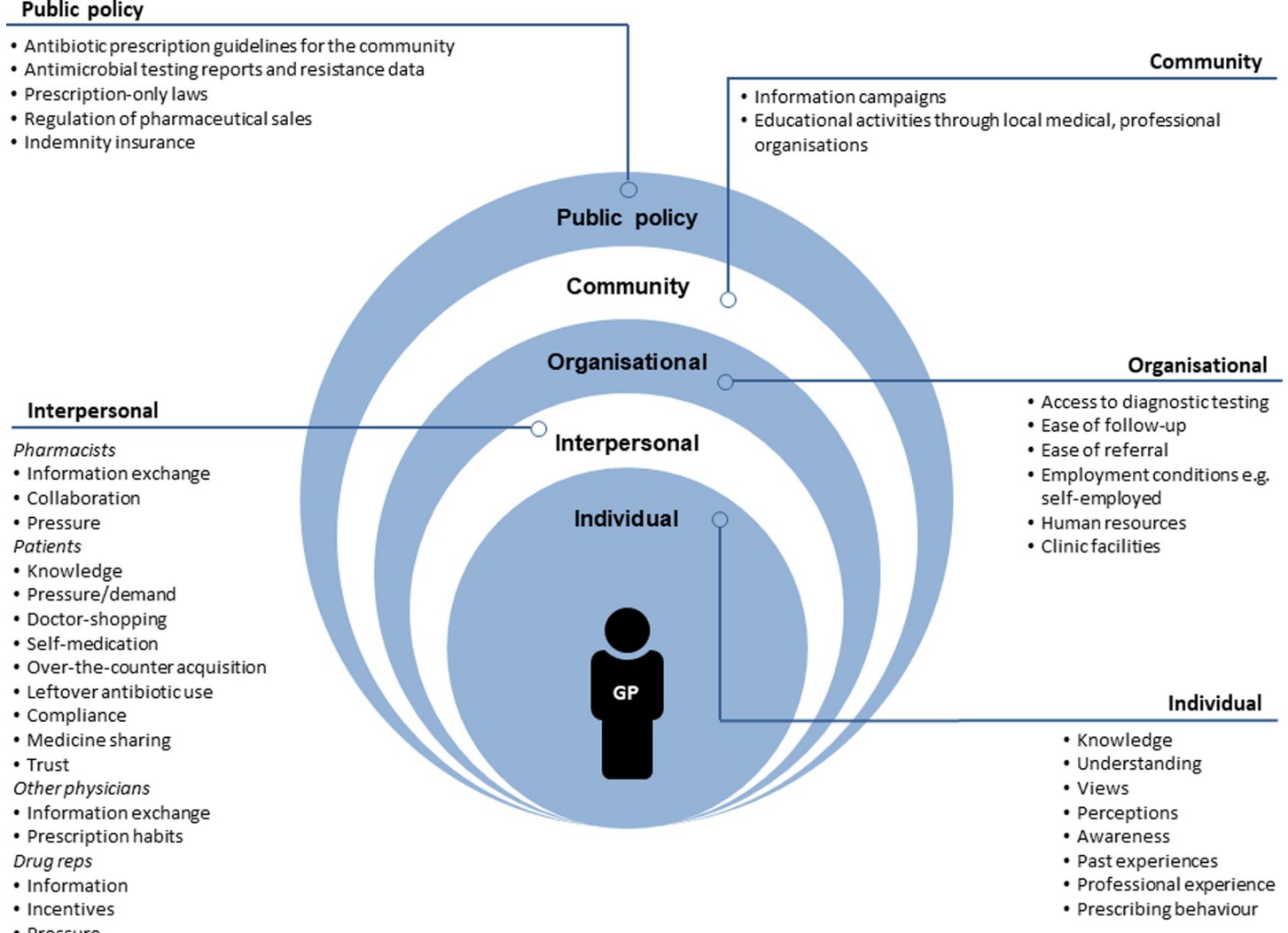

**Fig 1. Overview of barriers and facilitators identified to have an impact on GPs' antibiotic prescribing at all levels of the socioecological model.**

requesting patients call back or return for follow-up should they deteriorate, and delayed antibiotic prescription, were often perceived as good strategies to minimise antibiotic use, although seldom with the intention of limiting ABR.

> "Antibiotics are obviously very important, very special and very precious drugs. Why? Because we are running out of them. . . . They are wonder drugs, you cure people, you save lives. We underestimate what we do because we're continuously saving lives with them. . ." (GP-J; male private GP; 41 years' experience).

GPs were aware that antibiotic misuse accelerates ABR, restricting treatment options if left uncontrolled. Many believed that ABR is on the rise in Malta because of antibiotic overuse and misuse (particularly for viral infections) by prescribers, pharmacists, and the public alike. GPs remarked that broad-spectrum antibiotics are overused and that patients self-medicate, are non-compliant, retain leftover antibiotics, and acquire antibiotics over the counter, although

this practice has allegedly almost ceased following revision in prescription policies. Marketing of specific antibiotic classes by drug reps was also considered a driver.

> "...in the community you can basically touch resistance. Often you prescribe something that used to work, and patients return saying it hasn't worked." (GP-I; male private GP; 39 years' experience).

Few GPs opined that ABR directly impacted their practice. Rather, several GPs reported achieving good results when prescribing antibiotics although they noted that ABR forced them to prescribe more broad-spectrum antibiotics. GPs disagreed about antibiotic doses; some believed high doses are unnecessary, others favoured more aggressive antibiotic therapy to avoid resistance development. Seldom, GPs reported prescribing combination antibiotic therapy to ensure infection resolution or limit ABR development, although many believed this unnecessary.

Although GPs considered the majority (70–90%) of upper ARTIs to be viral, GPs found it difficult to determine their aetiology based on clinical presentation, particularly lower ARTIs. Some determined aetiology over time, typically waiting 2–5 days for an infection to resolve. GPs also often believed viral infections commonly progress to secondary bacterial infections, particularly in the elderly.

> "There is a tendency for old, frail patients with viral infections to get secondary bacterial infections... It would be indicated that it is a viral infection but being an old patient, prescribing an antibiotic won't do any harm." (GP-A; male private GP; 30 years' experience).

Infections non-responsive to antibiotic therapy were often believed to be antibiotic-resistant or progression from a viral infection to a secondary bacterial infection; few noted non-response could be due to antibiotics prescribed unnecessarily for a viral infection. Determining the cause of a non-responsive infection was perceived as challenging, particularly without local data, and sometimes resulted in the prescription of multiple antibiotic courses.

> "It's very difficult to know whether an infection is resistant or whether they are non-responsive to treatment because they never needed antibiotics in the first place." (GP-B; female public GP; 25 years' experience).

**Category II: GPs' self-reported antibiotic prescribing practices.**    Antibiotic prescribing was viewed by several GPs as their sole responsibility, however self-reported practices varied widely. GPs not in favour of prescribing antibiotics immediately often chose to delay antibiotic prescription, particularly in mild or suspected viral cases. They typically prescribed symptomatic relief and requested patients follow-up should they develop worsening or changing symptoms (typically within 5 days). If follow-up was not possible, GPs were more likely to prescribe antibiotics earlier.

> "A viral illness will last five days. I prefer to wait, and if there is an increase in fever, a change in the symptoms, worsening cough, greenish phlegm, you're going to start them on antibiotics at that stage." (GP-G; female public GP; 19 years' experience).

GPs acknowledged occasionally prescribing antibiotics without a focus of infection if in doubt or if they suspect the patient may deteriorate, safeguarding the patient and themselves. Some GPs described instances where patients deteriorated, were hospitalised, or died, and they themselves or their colleagues were accused of not prescribing antibiotics immediately.

Consequently, fear of missing possible life-threatening illnesses led them to err on the side of caution, even if not always in line with the evidence base.

"I had an adult . . . I prescribed antibiotics and it turned out to be a virus . . . I still gave antibiotics . . . if I missed the boat this person would end up with either a rupture or hearing loss so you can't afford not to give and then be blamed for not recognizing her condition. It is better to be on the safe side, both for myself and the patient." (GP-M; male public/private GP; 23 years' experience).

GPs considered several factors before prescribing antibiotics, including the patient's smoking status, age, occupation, and co-morbidities. Several believed smokers more often present with bacterial infections and children often acquire viral infections. Elderly patients and patients with multiple co-morbidities, were more likely to require antibiotics early according to GPs, to avoid complications and hospitalisation.

". . .often . . . I end up giving prophylactic antibiotics to 80-year-old patients because if you don't you end up with a chest infection which is harder to treat. These are people who easily end up hospitalised. . ." (GP-K; male public/private GP; 19 years' experience).

GPs often adjusted prescriptions to suit patients' needs and preferences, to improve compliance. Antibiotic allergies or discomforting side effects (e.g., thrush, nausea/vomiting, diarrhoea) typically led GPs to tailor therapy, changing antibiotic class if necessary. Some GPs also considered flavour (particularly in children), dosage form and regimen, and preferred antibiotics with daily or bidaily regimens.

"Some patients have a real hard time swallowing large tablets and prefer smaller tablets. Compliance is really dependent on these small details." (GP-F; female private GP; 25 years' experience).

**Category III: GPs' views on diagnostic testing and its availability in Malta.** Few GPs used diagnostic testing (e.g., X-rays, CRP tests, throat swabs, sputum cultures) unless they considered the patient's condition complicated. Public GPs were more willing to utilise such services due to their availability through the national health service; private GPs often lacked access to diagnostic facilities. Very few GPs were familiar with rapid POCTs. In fact, rapid POCTs for ARTIs were largely unavailable and almost all GPs had never used them. Few private GPs had access to rapid strep tests and only one GP used them regularly. Nevertheless, several GPs expressed positive attitudes towards them and were willing to test them if reliable and time efficient. They believed rapid POCTs could decrease antibiotic overuse and help patients understand their condition and treatment plan. Others insisted they would unlikely rely on rapid POCTs; they would rather rely on their experience and clinical assessment. They perceived rapid POCTs as tools used when unsure of oneself.

"They [rapid tests] would absolutely be very helpful. . . . You can explain to patients that the results are positive which means you have a bacterial infection. . . It can be used to persuade the patient rather than basing it just on my clinical impression." (GP-M; male public/private GP; 23 years' experience).

Time was considered a barrier to diagnostic testing, including delays caused by referrals and receipt of test results. For rapid POCTs, GPs' felt they lacked the necessary human resources

and infrastructure, which would cause time delays, negatively impacting their practice. Cost was another deterrent among many private GPs. Many believed patients would be unwilling to pay, and that they cannot increase consultation costs as a result. Some believed they may lose clients who will seek care elsewhere instead. This combination of barriers and limited access to testing, often led GPs to prescribe antibiotics empirically, occasionally prescribing a second antibiotic if the first fails to respond, before referring the patient for further investigation.

"We prescribe antibiotics without carrying out bacteriological tests, so we use our clinical judgement to try form an idea of what the underlying bacteriology is and treat it accordingly with the best antibiotic to match the infection." (GP-J; male private GP; 41 years' experience).

## Sub-theme II: Addressing patient behaviour, knowledge, and awareness through education

**Category IV: GPs' views on patient behaviour, knowledge, and awareness.**   Patient expectations, pressure and demand for antibiotics, non-compliance, and self-medication, were dominant problems reported by GPs. According to GPs, patients seek medical care upon onset of symptoms, even for self-limiting infections, and expect immediate cure.

". . .if a patient woke up with a bad sore throat he'll be seen immediately, and because of this, patients expect an immediate cure. You cannot tell them to come back in three days' time, they simply wouldn't grasp that." (GP-O; male private GP; 26 years' experience).

Some GPs found demanding patients impossible to persuade and reported that some patients express dissatisfaction when they do not get their way, sometimes threatening to consult another GP. This was viewed as a problem as GPs could lose patients and income as a result. Most GPs, however, felt largely unaffected by this behaviour. Experience was regarded an asset, giving GPs confidence to address such behaviour. Few GPs admitted to buckling under pressure occasionally, even when antibiotics are unnecessary. Some believed this to be harmless; others prescribed short courses, narrow-spectrum antibiotics, or delayed antibiotic prescriptions to reduce harm.

". . .when I fail to convince patients, because they literally do not understand, after a lot of thought I figure. . . one out of ten won't make . . . a difference." (GP-G; female public GP; 19 years' experience).

Some GPs felt that through increased awareness, antibiotic demand is decreasing, particularly among parents. Nonetheless, several GPs believed patients still harbour misconceptions including that fever requires antibiotics and antibiotics are anti-inflammatory drugs. According to GPs, many do not understand how to use antibiotics appropriately, cannot differentiate between viral and bacterial infections, and believe they cannot recover without antibiotics. Another reported misconception was that the body develops resistance to antibiotics.

**Category V: GPs' experiences with patient education.**   GPs reported dedicating lots of time towards patient education, viewing it as their responsibility. GPs typically informed patients about their diagnosis, disease progression, and symptom management. GPs also explained the importance of not overusing antibiotics, storing leftovers, self-medicating, or stopping treatment prematurely. Some tried to clarify how resistance develops and its repercussions, but considered it challenging. Although many successfully rationalised with

persistent patients why antibiotics are unnecessary, some felt it was pointless when patients insist on getting antibiotics against their advice.

### Sub-theme III: The role of other professionals

**Category VI: Interpersonal dynamics between GPs and other healthcare professionals.** GPs mentioned the importance of seeking advice from other physicians (GPs and other specialists) when necessary. Whilst GPs in health centres could rely on colleagues for guidance, some solo private GPs felt isolated. For many solo private GPs who rented clinics within pharmacies, working closely with pharmacists was considered advantageous. They appreciated having access to another healthcare professional to discuss treatment plans and request advice. Conversely, few GPs mentioned that pharmacists sometimes made GPs feel obliged to prescribe medicines.

"Some pharmacists give you horrible looks if the patient doesn't leave the clinic with a prescription." (GP-I; male private GP; 39 years' experience).

**Category VII: GPs' interaction with drug reps.** GPs frequently interacted with drug reps but had opposing views towards them and their influence on antibiotic prescription. Drug reps were viewed by many as important contact points, particularly in private practice. To these GPs, the educational and informative aspect of drug-repping was imperative, keeping them abreast with recent medical advancements. They were the GP's few sources of information on ABR patterns, new drugs, and technologies.

Whilst many GPs appreciated drug reps as credible sources of information, others were sceptical. They believed that information is not always scientifically sound, biased, and sometimes fabricated, since drug reps are under pressure to boost sales. They mentioned that recommendations are based on foreign evidence, and that over the years, drug reps have recommended specific broad-spectrum antibiotics and larger doses as a strategy to prevent ABR. GPs were also wary of certain marketing strategies, e.g., lectures organised and/or sponsored by pharmaceutical companies, incentives, and promoting antibiotics for their anti-inflammatory effect. Several GPs reported pressure from drug reps to prescribe although most insisted it did not influence them.

"I do not think medical reps affect me much. I . . . take in all the information . . . but the final decision must be my own. . ." (GP-F; female private GP; 25 years' experience).

### Sub-theme IV: Primary care organisation, and the impact of public policy and guidelines on GPs' antibiotic prescribing

**Category VIII: Primary care organisation.** GPs encountered challenges with the healthcare organisation, particularly communication among the primary, secondary, and tertiary sectors. Lack of IT infrastructure resulted in no electronic link with other healthcare providers, often delaying feedback from specialists and hospitals regarding referrals. These delays impacted continuity-of-care and frustrated GPs, deterring them from referring patients. Ensuring continuity-of-care was considered a very important aspect of patient care to many private GPs. Episodic care was believed to lead to a more defensive approach and possibly antibiotic over-prescription.

**Category IX: Public policy and guidelines.** Few GPs believed that the introduction of professional indemnity insurance in 2014, may have rendered more defensive prescribing practices, i.e., prescribing antibiotics to safeguard oneself from legal implications. Regarding

availability of community antibiotic prescribing guidelines and data on local ABR rates, reports were mixed. Whilst public GPs accessed information online, few private GPs seemed aware of this resource; many claimed to have no access to guidelines, despite their availability on the National Antibiotic Committee's webpage. Several GPs believed in the importance of following guidelines to support diagnosis and treatment, particularly in complicated cases, yet only two GPs actively looked up guidelines; others used them as a last resort. Seldom, GPs felt overburdened and were cynical towards guidelines, placing more value in their experience.

> "Often when you reach my age, guidelines just make you laugh. If you want a frank and honest answer, I do not abide by guidelines. Sometimes they are stupid." (GP-I; male private GP; 39 years' experience).

GPs noted that ABR data were outdated, and based on inpatient or foreign data, making them less relevant for community practitioners. Several expressed a desire for current, local community data to guide prescribing guidelines and improve GPs' trust in them. The lack of current ABR data and local guidelines was considered a major pitfall, leading GPs to prescribe more cautiously or aggressively. One GP insisted however, that antibiotics are not abused, rather overprescribed unintentionally; the lack of current local data forces GPs to prescribe antibiotics blindly, without knowing what drug to prescribe or at what dose. Consequently, GPs avoid risks, opting for antibiotics they know are safe and effective.

## Discussion

To our knowledge, this is the first qualitative study conducted in Malta that identifies key barriers and facilitators to GPs' antibiotic prescribing to inform the implementation of a tailored social marketing intervention. We identified numerous barriers and facilitators to appropriate antibiotic prescription. These were visualised and summarised through a socioecological model to depict the complexity of antibiotic prescribing. Such models can help address barriers and guide public health practice [36]. A recent systematic review and meta-ethnography of antibiotic prescribing for ARTIs by primary care professionals, highlighted an array of individual, interpersonal, and contextual-level influences on antibiotic prescription [37]. It was concluded that interventions must be context-specific and consider prescribers' perceived roles and priorities to be accepted and effective [37], further stressing the importance of this study. Below we discuss key findings and suggest possible factors that could be targeted to develop effective interventions to promote prudent antibiotic prescription in Malta and similar contexts.

### Individual-level factors: GP awareness and misconceptions

Although many GPs were aware that antibiotics are important drugs that should be used appropriately as misuse promotes ABR, GPs' reports revealed some misconceptions on the pathogenesis of ARTIs that must be addressed. GPs' self-reported behaviour also indicated that antibiotics, particularly broad-spectrum, are being overprescribed, as previously suggested [23–26]. Like other studies [38,39], ABR was considered an important and growing problem both locally and abroad, yet GPs believed it seldom impacted them. ABR was attributed to rampant and repeated antibiotic overuse and misuse, particularly broad-spectrum antibiotics. This heightened awareness on ABR may however negatively impact prescribing and lead GPs to prescribe more broad-spectrum antibiotics [39].

Of concern, GPs seldom believed that non-response to treatment could be due to unnecessary antibiotic prescription for a viral infection. This would be the more likely scenario, given

that upper ARTIs are typically viral and self-limiting [40]. Instead, a predominant view was that non-response to treatment was a consequence of a secondary bacterial infection. Although this phenomenon is possible [41], it appears to be over-estimated by GPs. This is especially relevant in Malta, where patients tend to consult GPs early in the course of their illness [26]. GPs typically waited at most five days before considering an infection to have developed into a bacterial infection. However, most uncomplicated viral ARTIs last around five to seven days and peak in severity between days three and six [42]. For self-limiting ARTIs such as bronchitis and sinusitis, symptoms can last around three weeks (sometimes longer) without antibiotics [43], therefore prescribing antibiotics this early is likely premature.

Similar to previous findings [44–49], GPs were more likely to prescribe antibiotics to the elderly and less likely to children. GPs believed that the elderly are more likely to deteriorate as a result of complications and secondary bacterial infections, whilst children are more likely to experience viral infections. GPs also believed smokers are more likely to deteriorate without antibiotics and were more inclined to prescribe antibiotics to them, corroborating our previous findings [44]. Although there is no evidence that antibiotics improve clinical outcomes in smokers [50], being a smoker has been found to be associated with antibiotic prescription in several studies [44,50–52].

### Interpersonal-level factors: Interaction with other key stakeholders

**Patient demand and expectations.**    GPs experience and are influenced by patient pressure and demand to prescribe antibiotics, which could result in inappropriate antibiotic prescription [46,53–57]. In our recent study, we found that GPs in Malta are more likely to prescribe antibiotics to patients who request them [44], as corroborated by some GPs in this study. GPs believed that patient demand is driven by their expectation for a "quick fix" and belief that antibiotics are the solution, as has been found elsewhere [53]. GPs also believed that some patients lack the necessary knowledge to understand why antibiotics are not always necessary, making them harder to persuade. Indeed, the public's knowledge and awareness on appropriate antibiotic use remains low in Malta, lower than EU average [23].

Patient expectations and demands are often over-estimated by physicians however, and early expectation-setting is critical to meet patient needs and ensure satisfaction [58]. GPs often mentioned that dissatisfied patients sometimes threaten to consult elsewhere. Since many patients pay out-of-pocket, GPs may feel obliged to prescribe antibiotics to avoid losing clients, avoid reconsultation, and/or ensure patient satisfaction [59]. However, giving patients an antibiotic prescription does not necessarily guarantee satisfaction. Whilst some studies have shown that an antibiotic prescription improves patient satisfaction [54,60], others have indicated that proper examination and information (without antibiotics) leads to higher satisfaction [61–64]. In fact, communication is sometimes valued more by patients than a prescription [65]. Given GPs' interest in patient education in this setting, introducing educational tools, like images and charts, to support GPs, could be beneficial.

**Influence of drug reps.**    Research shows that drug reps frequently visit GPs, providing one-on-one information during outreach visits [66–68]. Despite most GPs claiming to be unaffected by drug reps, as corroborated by previous studies [69], exposure to information from pharmaceutical companies can negatively impact GPs' quality of antibiotic prescribing. It could not only augment prescribing but also irrational prescription of the company's drug [66,68–71]. Such influences can potentially explain the predominance of broad-spectrum antibiotics in Malta in preference to narrower-spectrum alternatives, despite the former not being indicated epidemiologically [72]. It is relevant to highlight, that the proportion of drug reps to doctors in Malta is one of the highest in the EU. It is unlikely that local drug companies were willing to

expend so many resources if they did not perceive a satisfactory return on investment. There is no evidence that exposure to promotional activities by pharmaceutical companies improves antibiotic prescribing, therefore continued exposure cannot be recommended [66].

Yet, although some GPs received drug reps with scepticism in our study, GPs often felt that drug reps' educational and informative role was imperative, similar to other studies [69]. This was particularly prevalent in private solo practices where GPs experienced little interaction with other healthcare professionals. This demonstrates the need for alternate strategies, such as academic detailing, to disseminate scientifically-sound and evidence-informed guidelines. Similar to drug reps' tactics, in academic detailing, physicians receive individual educational visits by trained healthcare professionals within their own professional setting [73]. It gives GPs the opportunity to reflect on their prescription behaviour and learn about guideline-concordant recommendations for appropriate pharmacotherapy [55]. This can successfully decrease antibiotic prescription rates whilst improving quality and guideline-concordance [74,75]. Since GPs lack time to attend formal lectures, we believe that with enough resources, this strategy can bring correct and timely information to GPs in this type of healthcare setting.

## Organisational and public policy-level factors: Addressing diagnostic uncertainty

Lack of access to up-to-date national antibiotic guidelines and data on community ABR rates were considered major barriers to appropriate antibiotic prescription by GPs. Moreover, GPs lacked access to rapid POCTs and were often not familiar with them. GPs also acknowledged prescribing antibiotics without a definitive diagnosis out of fear of negative repercussions, and frequently expressed difficultly in differentiating between viral and bacterial infections. Diagnostic uncertainty is a major barrier to appropriate antibiotic prescription [53], that correlates with antibiotic misuse and overuse [14,15,76]. In cultures such as Malta, where uncertainty avoidance is a dominant cultural trait [77], we believe GPs could benefit from tools that help support decision-making, as discussed below.

**Antibiotic prescribing guidelines.**   Few GPs believed in the importance of antibiotic prescribing guidelines in this study. Whilst older GPs believed that clinical experience is superior to guidelines, as shown elsewhere [78], GPs were also sceptical about the accuracy of community antibiotic prescribing guidelines which, according to them, were based on foreign guidelines and hospital ABR rates. Lack of trust in guidelines is a major barrier to uptake [78,79]. To build trust, establishment of local community surveillance systems for antibiotic prescribing and ABR, coupled with timely data dissemination to all GPs is critically needed. Surveillance is one of the pillars of ABR containment [1,80,81] and demands urgent attention in this setting. Additionally, guidelines should consider patient needs and preferences, including side effects to specific antibiotics, co-morbidities, and polypharmacy [78,79]. Such factors often affected GPs' antibiotic prescribing decisions, therefore incorporating patient factors that influence GPs' likelihood to deviate from recommendations, may further improve adherence.

**Rapid POCTs.**   Rapid POCTs, such as point-of-care C-reactive protein testing, have been shown to reduce antibiotic prescribing rates cost-effectively [82]. When used appropriately, they can guide clinical management and mediate diagnostic certainty by providing GPs a better estimate of illness severity, particularly in settings where access to timely diagnostic services is limited. Rapid POCTs can also provide GPs with negotiating power and an educational opportunity with patients who insist on being prescribed antibiotics [83]. Consequently, GPs may succumb to patient demand less often [84,85], as shown in Spain, where POCTs decreased antibiotic prescription among patients who demanded antibiotics by 18.9% [84]. However, GPs in this study believed that introducing rapid POCTs may be impeded by time constraints,

added costs, lack of resources, and patient hesitancy. Despite an overall positive outlook towards rapid POCTs, some GPs would still rather rely on clinical assessment, expecting rapid POCTs to have little impact on their prescribing decisions, as reported elsewhere [83]. Nonetheless, we firmly believe that introducing rapid POCTs in Malta should be considered, although their utility will need to be rigorously assessed to ensure successful scale-up on a national level.

Pharmacists, who already perform rapid POCTs like urinalyses and blood glucose testing in Malta, could possibly collaborate in this strategy. Evidence suggests that involving pharmacists to detect GAS pharyngitis using rapid POCTs and providing Penicillin V treatment when necessary, is more cost-effective than treatment provided by doctors [86]. This approach could alleviate the burden of introducing rapid POCTs into general practices. However, given that several pharmacists run their own private pharmacies in Malta, allowing pharmacists to dispense antibiotics could be impacted by financial gain and should not be overlooked.

**Delayed antibiotic prescription.** Finally, delayed antibiotic prescribing strategies can also reduce antibiotic use for ARTIs by 45–80%, without significant negative patient repercussions [87–94]. GPs hold varying perceptions about delayed antibiotic prescription however, and there is variation in the way delayed antibiotic prescription strategies are used [95–97]. Whilst this variation cannot be disregarded, appropriate and targeted delayed antibiotic prescription strategies have a role to play in reducing unnecessary antibiotic use, particularly in high uncertainty avoidance cultures, providing reassurance to both the patient and GP [95,98–101]. Incorporating delayed antibiotic prescription into antibiotic prescribing guidelines is also recommended [95].

## Strengths and limitations

The strength of our study lies in the richness of our data, which allowed us to present our findings in a socioecological model where antibiotic prescribing is shown to be impacted not only by intrinsic GP knowledge and awareness, but a myriad of factors at all levels of the model. Although context-specific, our findings are relevant not only to local GPs but could also be important in other contexts. It is important to note however, that our data only captures GPs' views from 2014. Although still valuable and relevant given that little changes have occurred in this area in our context, further research should focus on better understanding and closely examining individual factors at all levels of the model, from various stakeholders' perspectives. This would help better understand and address factors that influence antibiotic prescription in a more comprehensive manner.

## Conclusion and recommendations

We identified key issues that must be addressed to successfully improve GPs' antibiotic prescribing behaviour for ARTIs. Through a socioecological model, we visualised the problem's complexity and the numerous influences on GPs' antibiotic prescribing decisions and behaviour. Future local initiatives must combine efforts across all levels to change behaviour and address misconceptions, whilst also improving the physical and social environment. In particular, GPs need access to relevant antibiotic guidelines for the community and more training and education with a particular focus on guidelines and enhancing GP-patient communication. Finally, more research is needed to better understand the general public's and pharmacists' views and behaviour, to design appropriate intervention strategies across all key stakeholders.

## Supporting information

**S1 File.**
(DOCX)

## Acknowledgments

We would like to express our immense gratitude to all GPs, for setting aside time from their hectic schedules to participate in this study.

## Author Contributions

**Conceptualization:** Erika A. Saliba-Gustafsson, Anna Nyberg, Michael A. Borg, Senia Rosales-Klintz, Cecilia Stålsby Lundborg.

**Data curation:** Erika A. Saliba-Gustafsson.

**Formal analysis:** Erika A. Saliba-Gustafsson, Anna Nyberg.

**Funding acquisition:** Erika A. Saliba-Gustafsson, Anna Nyberg, Michael A. Borg, Senia Rosales-Klintz, Cecilia Stålsby Lundborg.

**Investigation:** Erika A. Saliba-Gustafsson.

**Methodology:** Erika A. Saliba-Gustafsson, Anna Nyberg, Michael A. Borg, Senia Rosales-Klintz, Cecilia Stålsby Lundborg.

**Project administration:** Erika A. Saliba-Gustafsson.

**Resources:** Cecilia Stålsby Lundborg.

**Supervision:** Anna Nyberg, Michael A. Borg, Senia Rosales-Klintz, Cecilia Stålsby Lundborg.

**Visualization:** Erika A. Saliba-Gustafsson.

**Writing – original draft:** Erika A. Saliba-Gustafsson.

**Writing – review & editing:** Erika A. Saliba-Gustafsson, Anna Nyberg, Michael A. Borg, Senia Rosales-Klintz, Cecilia Stålsby Lundborg.

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
