## [Decision Letter · Decision Letter 0]

9 Sep 2020

PONE-D-20-11324

General practitioners’ understanding of antibiotic use and resistance, and perceived barriers and facilitators to prudent antibiotic prescribing: a qualitative study

PLOS ONE

Dear Dr. Saliba-Gustafsson,

Thank you for submitting your manuscript to PLOS ONE. After careful consideration, we feel that it has merit but does not fully meet PLOS ONE’s publication criteria as it currently stands. Therefore, we invite you to submit a revised version of the manuscript that addresses the points raised during the review process.

The two reviewers raised a couple of concerns that you should consider in a revised version (see end of mail). By the way, you will find most of the comments of Reviewer 2 in an additional file.

I also have some concerns and additional suggestions you which you will find also at the end of this mail.

We look forward to receiving your revised manuscript.

Kind regards,

Wolfgang Himmel

Academic Editor

PLOS ONE

Journal Requirements:

2.Thank you for stating the following in the Competing Interests section:

[I have read the journal's policy and the authors of this manuscript have the following competing interests: At the time of the study, SRK was employed at Karolinska Institutet, Sweden. She is currently employed by the European Centre for Disease Prevention and Control (ECDC). The views and opinions expressed herein are the authors’ own and do not necessarily state or reflect those of ECDC. ECDC is not responsible for the data and information collation and analysis and cannot be held liable for conclusions or opinions drawn. EASG, AN, MAB and CSL have no competing interests to declare.].

3.We note that you have indicated that data from this study are available upon request. PLOS only allows data to be available upon request if there are legal or ethical restrictions on sharing data publicly. For information on unacceptable data access restrictions, please see http://journals.plos.org/plosone/s/data-availability#loc-unacceptable-data-access-restrictions.

4. Please include a caption for figure 1.

Additional Editor Comments:

A real strength of your paper is the ecological model, or say, the parallel consideration of the many influences on a personal, professional, organizational level and so on. And exactly this makes the paper attractive not only for GPs in Malta but also in other countries, even in those with a lower use of antibiotics. Therefore, I would recommend that you make clearer than to date that the method of the paper and its results could also be of interest for readers worldwide.

Similar to Reviewer 2, I would also recommend that you start with the ecological model directly at the beginning of the Results section so that readers get a good overview of what they can expect on the following pages and how you organized your analysis and presentation of results.

Like Reviewer 1, I recommend a significant shortening of the manuscript. Even if PlosOne is an online journal, we should consider the readers interest to read concise manuscripts (“to the point”). Medical journals, even those with a qualitative focus (such as Family Practice) have usually word limits around 3,000 to 4,000 words. As far as I see, your manuscript has more than 10,000 words (!) and I ask you to limit it by a maximum of 6,000 words (fewer words would be even better!). You will see some suggestions later in my comments that may help to reach this limit.

The title of the manuscript, although rather long, misses and important term: “respiratory tract infections”. A shorter version, including this term could be:

Barriers and facilitators to prudent antibiotic prescribing in respiratory tract infections – a qualitative interview study with general practitioners in Malta (or “acute resp. …,” as often used)

I’m sure you find alternatives if you don’t agree with my suggestion.

Although I like the ecological model that guided, or helped to organize, your analysis, I have some concerns with the definitions and contents of your sub-themes and categories. Of course, I don’t want to interfere in the analysis of your working group, but perhaps you may find some other definitions in one or another case or re-work some of the categories:

See, for example, the first sub-theme. This is more or less the title of the paper or what you want to study but not a precise sub-theme. Do you find a more appropriate one?

Another example is a strong overlapping of category 1 and category 3 of the first sub-theme. Especially here, you could cancel a lot of text and concentrate on those aspects that are really outstanding and significant.

Also, sub-theme 1 and parts of sub-theme 2 are overlapping, especially as far as patient behavior is the issue. Again, here is room for strengthening the manuscript.

For me, the title of sub-theme 3 is too ‘positive’, if we have in mind, for example, drug reps. A better alternative may simply be: “The role of other professionals”.

One last example is category 9 of sub-theme 4. I feel parts of this category have nothing to do with ‘organization’ and so on but more or less with ‘attitudes’ of GPs towards testing and would then go well with sub-theme 1 (and could, again, be shortened!).

The Discussion is much too long and addresses too many issues, many of them already extensively discussed in the relevant literature. Please concentrate on the most important results from your research and the consequences for GPs and the international state of the art. Perhaps you can/may structure the discussion under 2 main issues: “GPs’ conceptions and misconceptions” and “external factors”. Just an idea.

For the Conclusion, I would recommend to restrict yourselves to 2 paragraphs: one with a stronger focus on the Malta GPs, one on the international discussion and future research.

As Reviewer 1, I also recommend to be precise when talking about patients and their behavior. Mostly, it’s not ‘patients’ but ‘patients and their behavior as perceived by the interviewees’. That is especially important in the case of ‘patient demand’, most often a matter overestimated by doctors but then ‘real in their consequences’. I think you know the rather old, but still excellent work of Paul Little and colleagues (https://www.bmj.com/content/bmj/328/7437/444.full.pdf) where we learn: “… after controlling for patient preference, medical need, and clustering by doctor, doctors' perceptions of patient pressure were strongly associated with prescribing … In all cases, doctors' perception of patient pressure was a stronger predictor than patients' preferences.”

Two minor remarks:

At several places in your paper, you talk about a ‘wait and see’ approach in such a way that one may think this is in some contrast to a ‘delayed prescription’ approach. Reading this interview with Geoffrey Spurling (https://medicalxpress.com/news/2017-09-dose-wait-and-see-unnecessary-antibiotic.html, I see no difference, also when reading his original paper (https://www.cochranelibrary.com/cdsr/doi/10.1002/14651858.CD004417.pub5/full?cookiesEnabled). By the way, you may cite the paper.

Maybe this paper, too, is worth to be referenced as an update of the Gulliford paper you cited:

https://pubmed.ncbi.nlm.nih.gov/30755451/

Reviewers' comments:

Reviewer's Responses to Questions

**Comments to the Author**

1. Is the manuscript technically sound, and do the data support the conclusions?

Reviewer #1: Yes

Reviewer #2: Yes

2. Has the statistical analysis been performed appropriately and rigorously? 

Reviewer #1: N/A

Reviewer #2: N/A

3. Have the authors made all data underlying the findings in their manuscript fully available?

Reviewer #1: No

Reviewer #2: Yes

4. Is the manuscript presented in an intelligible fashion and written in standard English?

Reviewer #1: Yes

Reviewer #2: Yes

5. Review Comments to the Author

Reviewer #1: The methods section could still use some additional information regarding sampling technique and also regarding the use of the socio ecological model.

Interview data is normally not made publically available as there is no consent for that given by the participants I assume

for further comments see attached document

Reviewer #2: The paper focuses on an important and timely health care issue. We read it with great interest. It is well structured and well written. Some aspects have already been described, but there are also new and interesting connections.

- line 118: „A target of 20 GPs was deemed necessary to ensure data saturation.“ – Is this a preliminary consideration? What does it result from? From my point of view, this is an unnecessary sentence, possibly delete it.

- line 160: EASG – that this is the author is not self-explanatory, I was thinking of a software. Please write e.g. „by autor EASG“ at the first time.

- It is written about RTI. This is very broad. Does it mean ARTI? Or can it be narrowed down if necessary?

- The results are very lengthy, maybe not everything has to be presented in every detail, please always concentrate on essentials. It is normal, that the categories can frequently not be separated accurately. Please avoid repeating yourself!

- The whole manuscript should be shortened, especially the results section. Here are some suggestions:

o Try to avoid overlapping passages: e.g. uncertainty of the GPs in line: 204 ff., 299-301 etc., 333-337) please in one section

o citation line 323-327 please delete (no new information)

o second citation line 403-406 please delete

o Category V, line 417-456: please cut, normal prescription behavior does not need to be described in every detail

o citation line 536-538 please delete

o citation line 566-570 please delete too

o line 583-589 please cut, e.g. „GPs often felt that they cannot afford to wait too long to treat patients as they can develop complications. Consequently they felt that they have no choice but to prescribe antibiotics without knowing whether they are truly warranted.“ could be deleted

o citation line 624-628 please delete, line 640-643, too

o line: 657 – 662 please cut (no new information)

o line 666-670: could be deleted

- The results regarding the GPs view on patients should be presented as such, that means: Please write about GPs perception of patients and not of patients as such. Your results are based on the way GPs experience and perceive their patients and are not based on patient views and reports themselves. This should be clear for the reader, e.g. “GPs perceive their patients to expect something…/ GPs have the feeling that their patients …. etc”.

6. PLOS authors have the option to publish the peer review history of their article (what does this mean?). If published, this will include your full peer review and any attached files.

Reviewer #1: No

Reviewer #2: No

---

## [Author Response · Author response to Decision Letter 0]

23 Oct 2020

Editor Comments:

1. Please include a caption for figure 1.

RESPONSE: Caption for figure 1 now added to the manuscript (lines 143-144).

2. A real strength of your paper is the ecological model, or say, the parallel consideration of the many influences on a personal, professional, organizational level and so on. And exactly this makes the paper attractive not only for GPs in Malta but also in other countries, even in those with a lower use of antibiotics. Therefore, I would recommend that you make clearer than to date that the method of the paper and its results could also be of interest for readers worldwide.

RESPONSE: Thank you for your positive remark. We have added a sentence on this in the methodological considerations section and have moved our socioecological model to the beginning of our findings to show case our results better.

3. Similar to Reviewer 2, I would also recommend that you start with the ecological model directly at the beginning of the Results section so that readers get a good overview of what they can expect on the following pages and how you organized your analysis and presentation of results.

RESPONSE: We took your advice and that of the reviewer and moved the socioecological model to the beginning of the results section.

4. Like Reviewer 1, I recommend a significant shortening of the manuscript. Even if PlosOne is an online journal, we should consider the readers interest to read concise manuscripts (“to the point”). Medical journals, even those with a qualitative focus (such as Family Practice) have usually word limits around 3,000 to 4,000 words. As far as I see, your manuscript has more than 10,000 words (!) and I ask you to limit it by a maximum of 6,000 words (fewer words would be even better!). You will see some suggestions later in my comments that may help to reach this limit.

RESPONSE: Thank you and the reviewers for suggesting areas where the word count can be reduced. We condensed the manuscript significantly. The current word count is: 6079.

5. The title of the manuscript, although rather long, misses an important term: “respiratory tract infections”. A shorter version, including this term could be: ‘Barriers and facilitators to prudent antibiotic prescribing in respiratory tract infections – a qualitative interview study with general practitioners in Malta’ (or “acute resp. …,” as often used). I’m sure you find alternatives if you don’t agree with my suggestion.

RESPONSE: Thank you for this suggestion. We took it into consideration and re-worded the title similar to the one suggested.

6. Although I like the ecological model that guided, or helped to organize, your analysis, I have some concerns with the definitions and contents of your sub-themes and categories. Of course, I don’t want to interfere in the analysis of your working group, but perhaps you may find some other definitions in one or another case or re-work some of the categories:

• See, for example, the first sub-theme. This is more or less the title of the paper or what you want to study but not a precise sub-theme. Do you find a more appropriate one?

RESPONSE: Thank you for this comment. We changed the title as per your previous suggestion and have also rephrased the name of sub-theme 1. It now reads, “GPs’ views on antibiotic use and resistance, and their management of ARTIs”.

• Also, sub-theme 1 and parts of sub-theme 2 are overlapping, especially as far as patient behavior is the issue. Again, here is room for strengthening the manuscript.

• For me, the title of sub-theme 3 is too ‘positive’, if we have in mind, for example, drug reps. A better alternative may simply be: “The role of other professionals”.

RESPONSE: Thank you for pointing this out to us. We agree with you and have decided to rename the sub-theme to “The role of other professionals” as suggested.

• Another example is a strong overlapping of category 1 and category 3 of the first sub-theme. Especially here, you could cancel a lot of text and concentrate on those aspects that are really outstanding and significant.

One last example is category 9 of sub-theme 4. I feel parts of this category have nothing to do with ‘organization’ and so on but more or less with ‘attitudes’ of GPs towards testing and would then go well with sub-theme 1 (and could, again, be shortened!).

RESPONSE: After considering these very important points you raised, we decided to reorganize our findings, collapsing categories 1 and 3, and moving category 9 to sub-theme 1 instead of sub-theme 4. We also considered your earlier point on avoiding overlap between sub-themes 1 and 2, and editing/re-arranged our results accordingly.

7. The Discussion is much too long and addresses too many issues, many of them already extensively discussed in the relevant literature. Please concentrate on the most important results from your research and the consequences for GPs and the international state of the art. Perhaps you can/may structure the discussion under 2 main issues: “GPs’ conceptions and misconceptions” and “external factors”. Just an idea.

RESPONSE: Thank you for this comment and your suggestion. We condensed the discussion considerably and rearranged the flow, including less sub-headings (lines 380-521).

8. For the Conclusion, I would recommend to restrict yourselves to 2 paragraphs: one with a stronger focus on the Malta GPs, one on the international discussion and future research.

9. As Reviewer 1, I also recommend to be precise when talking about patients and their behavior. Mostly, it’s not ‘patients’ but ‘patients and their behavior as perceived by the interviewees’. That is especially important in the case of ‘patient demand’, most often a matter overestimated by doctors but then ‘real in their consequences’. I think you know the rather old, but still excellent work of Paul Little and colleagues (https://www.bmj.com/content/bmj/328/7437/444.full.pdf) where we learn: “… after controlling for patient preference, medical need, and clustering by doctor, doctors' perceptions of patient pressure were strongly associated with prescribing … In all cases, doctors' perception of patient pressure was a stronger predictor than patients' preferences.”

RESPONSE: Thank you for drawing our attention towards this again. We have made the necessary changes through the manuscript to ensure that patient behaviours and attitudes reported are based on GPs’ own reports and perceptions. Thank you also for raising the findings by Paul Little and colleagues. They are very relevant and so we have cited this study in our discussion under the patient demand and expectations sub-section. 

Two minor remarks:

10. At several places in your paper, you talk about a ‘wait and see’ approach in such a way that one may think this is in some contrast to a ‘delayed prescription’ approach. Reading this interview with Geoffrey Spurling (https://medicalxpress.com/news/2017-09-dose-wait-and-see-unnecessary-antibiotic.html, I see no difference, also when reading his original paper (https://www.cochranelibrary.com/cdsr/doi/10.1002/14651858.CD004417.pub5/full?cookiesEnabled). By the way, you may cite the paper.

RESPONSE: We have cut down significantly on discussion regarding delayed antibiotic prescription strategies since we have already published another paper on GPs specific views on delayed antibiotic prescription and their practices. We would however like to continue to differentiate between the two approaches in our findings, as this is how it was described by GPs.

There is variation in the literature in definitions around delayed antibiotic prescription strategies. We believed that the strategy is described quite eloquently by McDermott et al (2017) (PMID: 29180593). They categorise it into four strategies (all with the intention of delayed antibiotics): (i) providing an antibiotic prescription and instructing patients to use it after a given timeframe if symptoms persist or worsen (patient-led strategy), (ii) post-dating (or forward-dating) the prescription so that patients cannot purchase it before a specified date, (iii) instructing patients to return to collect the prescription at a later date if needed, or (iv) requesting that the patient call the clinic/practitioner to issue a prescription should certain criteria be met.

11. Maybe this paper, too, is worth to be referenced as an update of the Gulliford paper you cited: https://pubmed.ncbi.nlm.nih.gov/30755451/

RESPONSE: Thank you for drawing this article to our attention. It has now been cited.

Reviewers' comments to the author:

Reviewer #1:

1. The methods section could still use some additional information regarding sampling technique and the use of the socioecological model.

RESPONSE: Please see lines 118-121 where information was added about the inclusion of the socioecological model to illustrate our results.

2. Interview data is normally not made publicly available as there is no consent for that given by the participants I assume.

RESPONSE: That is correct. Requests to make interview data publicly available were never made (was never the intention to do so) and so participants did not consent to that which is why we will refrain from making data publicly available, particularly since the data were collected in a small setting which could possibly make participants identifiable.

Methods

3. Line 28: ‘quota sample’, what does this mean? How does this differ from purposeful sampling and how does this then link to concept of data saturation?

RESPONSE: Quota sampling is a form of purposive sampling but basically first splits the target group of interest into categories that are deemed important. For us for example, we wanted to ensure variation in, (i) years of experience, (ii) sex, and (iii) locality of residence based on the theory that antibiotic prescribing might differ among these three factors of interest. We could not break this down into other factors of interest, e.g. public versus private GPs because we did not have enough data to do so during data collection. By breaking down our sampling into these three categories, we ensured variation in our sample for those three factors of interest that was also proportional to the population we sampled from. Data saturation does not have anything to do with this sampling approach. We initially had a target of 20 GPs as often it has been shown in literature that 20 respondents are sufficient to ensure data saturation when using content analysis. In fact we did find that we had already reached saturation by GP 15 but continued to interview five more GPs because the interview had already been set up at that point.

4. Same comment to line 118, what kind of sampling was used? Do you mean with quota sampling some form of purposeful sampling?

RESPONSE: Please see comment above. Too make the article more concise, we removed the information on data saturation.

5. The interview data is already rather old as the interviews were done in 2014. It does not mean the data is not valuable anymore, but it would be good to put it in context of that time in qualitative research especially in the discussion section and limitation. 

6. Line 149: had the interviewer some experience in qualitative research? Has she been trained in interviewing?

RESPONSE: Thank you for pointing this out. Yes, the first author has both training and experience in qualitative research. This has now been added to the methods (lines 106-107).

7. Line 164: what is the background of AN (as this might have an impact on the data analysis so it is good to know as a reader). Have both persons analysed all transcripts or only a few?

RESPONSE: AN has background in marketing and economy, and experience in qualitative research. Her insights brought an added dimension to the analysis which formed part of a larger project that utilised social marketing methods to identify key barriers and facilitators to antibiotic prescribing, to subsequently developed a targeted intervention, specifically addressing needs identified in this study. We added a short description on lines 112-114.

Results but linked with methods

There is an elaborate description of the many subthemes and then at the end there is a small description of the socioecological model, I think in my personal opinion, as I was reading through the results I was already thinking that applying that model would have been useful on the dataset. You do that at the end but maybe it would make the results section better if you organize it in that way from the beginning and explain this also in the method section, it will make the piece a lot stronger and easier to read.

RESPONSE: Thank you for sharing your views on this. As recommended, we shifted the results to start with the model. We also added a sentence in the data analysis section in the methods to describe how we used the model to illustrate our findings across all levels of the model (please see lines 119-122).

Discussion

In my view, the description of the results again could be shorted a bit or maybe interpreted a little bit more as it is now a bit repetitive of the results section. Maybe it could be useful to start from the socioecological model and link to the most important point that will need to be addressed in interventions. Also highlight what is new in your research compared to other research or what is more specific for Malta (as this is the reason why this research has been set up) than what you can find in the literature.

Methodological considerations

Starting from Line 908

“In qualitative research it is customary to use small, non-probability samples. We used quota sampling, a strategy that considers sizes and proportions of subsamples, in order to obtain subgroups that reflect corresponding proportions in the population and therefore attempt to gain a more representative picture”

I would not start with defending the ‘small sample’ in qualitative research this is normal and we shouldn’t defend it but it is important to be reflective about your sample, who have you sampled, who haven’t you got sampled, what might you have missed… to ensure quality of your data. Could you be a bit more reflective of your sample size, would there be potential participants that you might have missed? Data saturation is also linked to who you have selected for interviewing but also the interview questioning (the more structured the quicker you will get ‘saturation’ for example) but it is also linked to the experiences of the interviewer, how skilled are they? So it would be good to be a bit more reflective about that as well (and to mention also these skills in the method section) as these are (and some more) are essential criteria for saying we have data sufficiency.

I am not sure the aim in qualitative research is to get a representative picture for the whole population, but it is rather the aim is to find information rich cases to the phenomenon under investigation and personally I am more in favour of using the term ‘transferability’ to argue for the potential of wider relevance of the results which can be the result of the composition of the sample (i.e. diverse sample), rather than the sample size.

RESPONSE: Thank you for your comment. We have made edits to both the methods and methodological consideration sections based on your comments. 

Reviewer #2:

8. Line 118: target of 20 GPs was deemed necessary to ensure data saturation. Is this a preliminary consideration? What does it result from? From my point of view, this is an unnecessary sentence, possibly delete it.

RESPONSE: This was a preliminary consideration as it is often shown that 20 informants are sufficient to ensure data saturation when using content analysis. During data collection we reach saturation by about the 15th interview but continued with already scheduled interviews to reach 20. However, as you rightly point out, this detailed information is unnecessary therefore for word count purposes this sentence has now been deleted.

9. Line 160: EASG – that this is the author is not self-explanatory, I was thinking of a software. Please write e.g. by author EASG at the first time.

RESPONSE: This was first described in the first line of the data collection section. We have retained it to avoid confusion. Please see line 106.

10. It is written about RTI. This is very broad. Does it mean ARTI? Or can it be narrowed down if necessary?

RESPONSE: Thank you for pointing this out. We have changed RTI to ARTI throughout since we interviewed GPs specifically about acute respiratory tract infections (both upper and lower).

11. The results are very lengthy, maybe not everything has to be presented in every detail, please always concentrate on essentials. It is normal, that the categories can frequently not be separated accurately. Please avoid repeating yourself!

RESPONSE: Thank you for your comment and advice. We reorganised the results and made a great effort to reduce the overall word count by condensing and text. The word count has now been reduced to 6079words.

12. The whole manuscript should be shortened, especially the results section. Here are some suggestions:

• Try avoiding overlapping passages: e.g. uncertainty of the GPs in line: 204 ff., 299-301 etc., 333-337) please in one section

• citation line 323-327 please delete (no new information)

• second citation line 403-406 please delete

• Category V, line 417-456: please cut, normal prescription behaviour does not need to be described in every detail

• citation line 536-538 please delete

• citation line 566-570 please delete too

• line 583-589 please cut, e.g. “GPs often felt that they cannot afford to wait too long to treat patients as they can develop complications. Consequently, they felt that they have no choice but to prescribe antibiotics without knowing whether they are truly warranted.” could be deleted

• citation line 624-628 please delete, line 640-643, too

• line: 657 – 662 please cut (no new information)

• line 666-670: could be deleted

RESPONSE: Thank you for these helpful suggestions. We took them all into consideration and efforts were made to reduce the overall word count by condensing and rearranging the text. The word count has been reduced to 6079 words.

13. The results regarding the GPs view on patients should be presented as such, that means: Please write about GPs perception of patients and not of patients as such. Your results are based on the way GPs experience and perceive their patients and are not based on patient views and reports themselves. This should be clear for the reader, e.g. “GPs perceive their patients to expect something… GPs have the feeling that their patients …. etc”.

RESPONSE: Thank you for drawing our attention towards this. We have made the necessary changes through the manuscript to ensure that patient behaviours and attitudes reported are based on GPs’ own reports and perceptions.

---

## [Decision Letter · Decision Letter 1]

10 Dec 2020

PONE-D-20-11324R1

Barriers and facilitators to prudent antibiotic prescribing for acute respiratory tract infections: a qualitative study with general practitioners in Malta

PLOS ONE

Dear Dr. Saliba-Gustafsson,

Thank you for submitting your manuscript to PLOS ONE. After careful consideration, we feel that it has merit but does not fully meet PLOS ONE’s publication criteria as it currently stands. Therefore, we invite you to submit a revised version of the manuscript that addresses the points raised during the review process.

While Reviewer 2 accepted the manuscript as it is, Reviewer 1 has still some minor concerns (see below) which I ask to consider for the final version. 

I have also some minor concerns and a major concern which you hopefully may address for the final version:

I would recommend to move the last sentence of the Findings section in the Abstract (“Findings were collated … prescribing.”) to the end of the Methods section. And you should present 2 or 3 more results in the Findings section of the Abstract. I think you have room enough there and it makes the Abstract more interesting and this may stimulate readers to read the whole paper.

You talk about a “formative” study at the beginning of the Methods. I’m afraid this is an unusual description of your study type and/or irritating for readers since they typically know this expression form educational research (summative – formative assessment). Is “qualitative” not sufficient – what do you think?

A better heading of Category III may be: “Why GPs use or do not use diagnostic tests”. I think it’s not so much “attitudes” that matter but structure (e. g., access to facilities, lack of resources and so on). Of course, your decision!

Perhaps “GPs’ experience in patient education” is a better heading for Category V (instead of “role”).

Following Reviewer 1, my major concern refers to the structure of your Discussion section. Your socioecological model is really interesting but I was disappointed that you did not use this innovative approach to structure your discussion. Instead, you discuss several of your results, sometimes with a subheading, sometimes without (e.g., when discussing the role of pharmacists). For the reader, this selection of issues and the structure of how you present them must appear arbitrary. I strongly recommend to try the following: Take the subheadings of your socioecological model as subheadings of your Discussion and discuss (as short as possible!) how the interviews with the GPs inform this model or to put it the other way how the barriers and facilitators to prudent prescribing cannot be understood if we only look at GPs and their education/knowledge. The model helps us to consider the many other factors that influence antibiotic prescribing. I know that the first three areas or levels of the model are far better represented in your interview material than “community” and “public policy”, but, for example, guidelines and insurance do play a role in “public policy” for GPs, as you found out.

So, please try to structure the discussion this way and get back to me if it is impossible. If it is possible (and I do hope so!) you should report one important “limitation” of your study or study design. Even if the model helps to better understand antibiotic prescribing, your study only captures the model as seen/perceived by the GPs. Further research should examine the 5 areas of the model not alone from the GP perspective but as ‘realities’ of their own. Questions, for example, could be: Did information campaigns take place and did they change antibiotic prescriptions ad so on. I think your paper could show how the future of research in drug prescribing could be and what elements are needed. Take the chance!

I’m sure this revision will not take so much time as the last revision and I look forward to your final version.

We look forward to receiving your revised manuscript.

Kind regards,

Wolfgang Himmel

Academic Editor

PLOS ONE

Reviewers' comments:

Reviewer's Responses to Questions

**Comments to the Author**

1. If the authors have adequately addressed your comments raised in a previous round of review and you feel that this manuscript is now acceptable for publication, you may indicate that here to bypass the “Comments to the Author” section, enter your conflict of interest statement in the “Confidential to Editor” section, and submit your "Accept" recommendation.

Reviewer #1: (No Response)

Reviewer #2: All comments have been addressed

2. Is the manuscript technically sound, and do the data support the conclusions?

Reviewer #1: Yes

Reviewer #2: Yes

3. Has the statistical analysis been performed appropriately and rigorously? 

Reviewer #1: N/A

Reviewer #2: N/A

4. Have the authors made all data underlying the findings in their manuscript fully available?

Reviewer #1: Yes

Reviewer #2: Yes

5. Is the manuscript presented in an intelligible fashion and written in standard English?

Reviewer #1: Yes

Reviewer #2: Yes

6. Review Comments to the Author

Reviewer #1: dear authors,

well done on addressing comments adequately apart from comment 5 about the data been collected in 2014, so still valuable but it might already be a bit dated, so you need to address this in the strenght and limitation section.

The discussion still needs a bit of work I think, as you have used the socio ecological model I would expect to find some findings and implications related to those different levels (intrapersonal, interpersonal, organisation and community and policy level)

word count is still a lot, I personally find an article of more then 5000 words too lenghty and people might not read it. It is worth trying to get the essence said in under 5000 words

Reviewer #2: (No Response)

7. PLOS authors have the option to publish the peer review history of their article (what does this mean?). If published, this will include your full peer review and any attached files.

Reviewer #1: **Yes: **prof dr Sibyl Anthierens

Reviewer #2: No

---

## [Author Response · Author response to Decision Letter 1]

17 Jan 2021

Dear Dr Himmel,

Thank you for giving us the opportunity to revise our manuscript and for considering it publication in PLOS ONE. Thank you also to the you and the reviewer for taking the time to thoroughly critique our manuscript. Enclosed kindly find our responses to all comments received together with an updated version of our manuscript (both marked and unmarked). We made several new revisions to the manuscript, including restructuring the discussion and reducing the word count further.

We look forward to hearing back from you at your earliest convenience.

Kind regards,

ERIKA A. SALIBA-GUSTAFSSON, PHD

Academic Editor Comments:

1. I would recommend moving the last sentence of the Findings section in the Abstract (“Findings were collated … prescribing.”) to the end of the Methods section. And you should present 2 or 3 more results in the Findings section of the Abstract. I think you have room enough there and it makes the Abstract more interesting and this may stimulate readers to read the whole paper.

RESPONSE: We agree that it would be better to move the last sentence of our findings to the methods and have now done so. Thank you for pointing that out to us! We also presented additional findings in the abstract as recommended.

2. You talk about a “formative” study at the beginning of the Methods. I’m afraid this is an unusual description of your study type and/or irritating for readers since they typically know this expression form educational research (summative – formative assessment). Is “qualitative” not sufficient – what do you think?

RESPONSE: Thank you for pointing this out to us. To avoid any misunderstanding (and to further reduce the word count) we have now removed “formative” from the beginning of the Methods sections.

3. A better heading of Category III may be: “Why GPs use or do not use diagnostic tests”. I think it’s not so much “attitudes” that matter but structure (e. g., access to facilities, lack of resources and so on). Of course, your decision!

RESPONSE: Thank you for your comment. We have considered your recommendation and chose to revise the category to, “GPs’ views on diagnostic testing and its availability in Malta” (lines 252-253). We changed attitudes to views since the findings reflect GPs’ views of the situation in Malta, which is of course very much impacted by their lack of access to diagnostic testing facilities in this setting.

4. Perhaps “GPs’ experience in patient education” is a better heading for Category V (instead of “role”).

RESPONSE: Category V has now been rephrased to read “GPs’ experiences with patient education” (line 316).

5. Following Reviewer 1, my major concern refers to the structure of your Discussion section. Your socioecological model is really interesting but I was disappointed that you did not use this innovative approach to structure your discussion. Instead, you discuss several of your results, sometimes with a subheading, sometimes without (e.g., when discussing the role of pharmacists). For the reader, this selection of issues and the structure of how you present them must appear arbitrary. I strongly recommend to try the following: Take the subheadings of your socioecological model as subheadings of your Discussion and discuss (as short as possible!) how the interviews with the GPs inform this model or to put it the other way how the barriers and facilitators to prudent prescribing cannot be understood if we only look at GPs and their education/knowledge. The model helps us to consider the many other factors that influence antibiotic prescribing. I know that the first three areas or levels of the model are far better represented in your interview material than “community” and “public policy”, but, for example, guidelines and insurance do play a role in “public policy” for GPs, as you found out. So, please try to structure the discussion this way and get back to me if it is impossible.

RESPONSE: Thank you for your comment. We have taken your advice into consideration and restructured the discussion to highlight key issues whilst better reflecting the socioecological model. Since organisational-level and public policy-level factors have a great impact on diagnostic uncertainty, which is discussed in the paper, we chose to combine the two, splitting that sub-section into various headings relevant to those levels of the socioecological model.

6. If it is possible (and I do hope so!) you should report one important “limitation” of your study or study design. Even if the model helps to better understand antibiotic prescribing, your study only captures the model as seen/perceived by the GPs. Further research should examine the 5 areas of the model not alone from the GP perspective but as ‘realities’ of their own. Questions, for example, could be: Did information campaigns take place and did they change antibiotic prescriptions and so on. I think your paper could show how the future of research in drug prescribing could be and what elements are needed. Take the chance!

RESPONSE: Thank you for pointing this out. We have changed our methodological considerations to focus on strengths and limitations of the study, to include both limitations related to only including one perspective (the GPs’) but also to address reviewer #1’s comment (that the data dates to 2014). It is worth noting however, that little has changed since 2014 in this context, so our findings are still relevant.

Reviewers' Comments:

Reviewer #1:

Dear authors, well done on addressing comments adequately apart from comment 5 about the data been collected in 2014, so still valuable but it might already be a bit dated, so you need to address this in the strength and limitation section.

The discussion still needs a bit of work I think, as you have used the socioecological model, I would expect to find some findings and implications related to those different levels (intrapersonal, interpersonal, organisation and community and policy level) word count is still a lot, I personally find an article of more than 5000 words too lengthy and people might not read it. It is worth trying to get the essence said in under 5000 words.

RESPONSE: Thank you for your comments. We have done our utmost to address these concerns. Please see our responses below.

• We have included a strengths and limitations section that raises the issue of our data being collected in 2014.

• Discussion has been re-structured to reflect the socioecological model better.

• We noted your concern regarding the word count and reduced it further.

---

## [Editor Report · Decision Letter 2]

27 Jan 2021

Barriers and facilitators to prudent antibiotic prescribing for acute respiratory tract infections: a qualitative study with general practitioners in Malta

PONE-D-20-11324R2

Dear Dr. Saliba-Gustafsson,

We’re pleased to inform you that your manuscript has been judged scientifically suitable for publication and will be formally accepted for publication once it meets all outstanding technical requirements.

Kind regards,

Wolfgang Himmel

Academic Editor

PLOS ONE

Additional Editor Comments (optional):

Dear authors,

I would like to add that it was a pleasure to read both revisions and to see how the paper improved during the revisions so that is now a clearly written paper and will add to our knowledge. I do hope that you share my view although it was a lot of work for you (I appreciate it!).

By the way, when you submit the final version, you may consider to change the heading of the subchapter 'Organisational and public policy-level factors: addressing diagnostic uncertainty' simply into 'Addressing organisational, community and public policy factors'. Thus, you could include the 'community' level so that all levels of the Figure are addressed in the Discussion. And perhaps you may add a sentence somewhere in this section, such as: "Educational activities and information campaigns in the community could help to support the adequate prescription and use of antibiotics" (sorry for my English). Indeed, I think this is one message of the Figure and it is important not only to call on doctors but to include the community when implementing a prudent drug strategy.

But, of course, it is your decision whether or not you follow my suggestion.

Thanks, Wolfgang Himmel
---

## [Editor Report · Acceptance letter]

1 Feb 2021

PONE-D-20-11324R2 

Barriers and facilitators to prudent antibiotic prescribing for acute respiratory tract infections: a qualitative study with general practitioners in Malta 

Dear Dr. Saliba-Gustafsson:

I'm pleased to inform you that your manuscript has been deemed suitable for publication in PLOS ONE. Congratulations! Your manuscript is now with our production department. 

Kind regards, 

on behalf of

Professor Wolfgang Himmel 

Academic Editor

PLOS ONE